# Variation in the Caprine KAP24-1 Gene Affects Cashmere Fibre Diameter

**DOI:** 10.3390/ani9010015

**Published:** 2019-01-05

**Authors:** Jiqing Wang, Huitong Zhou, Yuzhu Luo, Mengli Zhao, Hua Gong, Zhiyun Hao, Jiang Hu, Jon G.H. Hickford

**Affiliations:** 1Gansu Key Laboratory of Herbivorous Animal Biotechnology, Faculty of Animal Science and Technology, Gansu Agricultural University, Lanzhou 730070, China; wangjq@gsau.edu.cn (J.W.); huitong.zhou@lincoln.ac.nz (H.Z.); luoyz@gsau.edu.cn (Y.L.); 18394187234@163.com (M.Z.); hua.gong@lincoln.ac.nz (H.G.); hzy18298352964@163.com (Z.H.); 2International Wool Research Institute, Faculty of Animal Science and Technology, Gansu Agricultural University, Lanzhou 730070, China; 3Gene-Marker Laboratory, Faculty of Agriculture and Life Sciences, Lincoln University, Lincoln 7647, New Zealand

**Keywords:** keratin associated protein 24-1 (KAP24-1) gene, variation, cashmere, fibre diameter, goat

## Abstract

**Simple Summary:**

The keratin-associated proteins (KAPs) are structural components of cashmere fibres. The human and sheep orthologous gene encoding high-sulphur (HS)-KAP24-1 is now described for the goat species. Our study identified the caprine KAP24-1 gene on goat chromosome 1, and we found that the gene was polymorphic and that variation in the gene affected cashmere fibre diameter.

**Abstract:**

The keratin-associated proteins (KAPs) are structural components of cashmere fibres. The gene encoding the high-sulphur (HS)-KAP24-1 (*KRTAP24-1*) has been identified in humans and sheep, but it has not been described in goats. In this study, we report the identification of caprine *KRTAP24-1*, describe variation in this gene, and investigate the effect of this variation on cashmere traits. A search for sequences orthologous to the ovine gene in the goat genome revealed a 774 bp open reading frame on chromosome 1, which could encode an HS-KAP. Based on this goat genome sequence and comparison with ovine *KRTAP24-1* sequences, polymerase chain reaction (PCR) primers were designed to amplify an 856 bp fragment that would contain the entire coding region of the putative caprine *KRTAP24-1*. Use of this PCR amplification with subsequent single-strand conformation polymorphism (SSCP) analysis of the amplicons identified four distinct patterns of DNA bands on gel electrophoresis, with these representing four different DNA sequences (*A* to *D*), in 340 Longdong cashmere goats reared in China. The variant sequences had the highest similarity to *KRTAP24-1* sequences from sheep and humans, suggesting that they are variants of caprine *KRTAP24-1*. Nine single-nucleotide polymorphisms (SNPs) were detected in the gene, including four non-synonymous SNPs and an SNP in proximity to the ATG start codon. Of the three common genotypes (*AA*, *AB*, and *BB*) found in these Longdong cashmere goats, cashmere fibres from goats of genotype *AA* had lower mean fibre diameter (MFD) than did those of genotype *AB*, and cashmere fibres from goats of genotype *AB* had lower MFD than did those from goats of genotype *BB*.

## 1. Introduction

The fleece of cashmere goats comprises cashmere, which is the fibre produced by the secondary fibre follicles, and higher-fibre-diameter “guard hairs” that are produced by the primary fibre follicles. As cashmere is a valuable product, it is separated by combing from the guard hairs, which are discarded, and the value of goat fleece is determined by particular cashmere fibre traits. The weight fleece weight and the mean fibre diameter (MFD) are the most valuable of the cashmere traits that can be measured. They, in large part, control the financial return to cashmere producers [1]. It is known that both genetic and environmental factors affect variation in cashmere traits. Accordingly, the identification of genes that regulate cashmere quantity and quality offers some opportunity to improve cashmere production.

Structurally, cashmere fibres are composed of keratins, which are assembled into intermediate filaments (IFs), and keratin-associated proteins (KAPs), which form a semirigid matrix that cross-links with the keratins in the IFs. The KAPs are therefore believed to play an important role in determining the physico-mechanical properties of the fibre.

The KAPs characteristically possess a high content of either cysteine or both glycine and tyrosine, and they can be classified into three broad groups based on their amino acid composition: the high-sulphur (HS; ≤30 mol.% cysteine) KAPs, the ultra-high-sulphur (UHS; >30 mol.% cysteine) KAPs, and the high-glycine/tyrosine (HGT; 35−60 mol.% glycine and tyrosine) KAPs [2]. The KAPs can then be further subdivided into families based on sequence similarity, and, to date, 27 paralogous KAP families have been identified across mammalian species [3].

The KAPs are encoded by small intron-less genes called *KRTAPs* [3]. While there are over 80 functional *KRTAPs* from 25 families identified in humans, and 29 *KRTAPs* from 13 families reported in sheep [3,4,5], only 12 *KRTAPs* from 9 families have been described in goats, including the recently identified *KRTAP20-2* [6,7]. Of these genes, variation in four (*KRTAP13-1*, *KRTAP8-2*, *KRTAP20-1*, and *KRTAP20-2*) has been described as being associated with cashmere fibre traits [7,8,9,10]. In sheep, where more studies have been carried out, additional *KRTAPs* have been found to be associated with wool traits, including *KRTAP1-2* [11], *KRTAP6-1* [12,13], *KRTAP6-3* [14], *KRTAP8-2* [15], *KRTAP22-1* [5], and *KRTAP26-1* [4]. This suggests that it would be worthwhile to identify new caprine *KRTAPs* and characterise variation in these genes, and then ascertain the effect of variation, should it exist, on cashmere fibre traits.

An HS-KAP gene named *KRTAP24-1* has been identified in humans [16] and in sheep [17]. Despite being a member of the HS group, the KAP24-1 protein encoded by human and sheep *KRTAP24-1* contains an unusually low level of cysteine when compared to other HS-KAPs [16,17]. Ovine *KRTAP24-1* is clustered with other HGT-*KRTAPs* and some HS-*KRTAPs* in a chromosome region where a number of associations with a variety of wool fibre traits have been reported [4,5,11,12,13,14,15], and similar findings have been reported in goats [7,8,9,10]. To date, the ovine *KRTAP24-1* orthologue has not been identified in goats, and its effect on fibre traits has not been reported in any other species. In this study, we report the identification of caprine *KRTAP24-1*, describe variation in this gene, and investigate the effect of this variation on cashmere traits.

## 2. Materials and Methods

### 2.1. Goats Investigated and Cashmere Data Collection

The animal experiments were carried out in accordance with the guidelines for the care and use of experimental animals established by the Ministry of Science and Technology of the People’s Republic of China (Approval Number 2006-398), and the work was approved by Gansu Agricultural University, Lanzhou, China.

In total, 340 Longdong cashmere goats were investigated. These were the progeny of ten unrelated sires. The goats were farmed by the Yusheng Cashmere Goat Breeding Company, which is located in Huan County of the Gansu Province of China. At one year of age (i.e., at first combing), the weight of cashmere fibre retrieved by combing was recorded. Fibres collected from the mid-side region of the goats were measured to ascertain the crimped fibre length and the mean fibre diameter (MFD), which was assessed by the Inner Mongolia Agricultural University, Inner Mongolia, China. Blood samples were collected onto Munktell TFN paper (Munktell Filter AB, Falun, Sweden), and DNA for PCR amplification was purified using a washing procedure described by Zhou et al. (2006) [18].

### 2.2. Search for the Caprine KAP24-1 Gene

A previously reported sheep *KRTAP24-1* sequence (GenBank accession no. JX112014) was used to BLAST search the Caprine Genome Assembly GCF_001704415.1 [19]. The sequence that shared the greatest similarity with JX112014 was assumed to be *KRTAP24-1*.

### 2.3. Polymerase Chain Reaction Single-Strand Conformation Polymorphism (PCR-SSCP) Analysis of Caprine KRTAP24-1

Based on the goat genome sequence identified above, and the comparison with sheep *KRTAP24-1* sequences (GenBank accession no. JX112014-JX112017), two PCR primers (5′- AGCCACAGTCTCGCCATAC-3′ and 5′- AGGTGGCACCTGCACCTTG-3′) were designed to amplify an 856 bp fragment that would contain the entire coding region of the putative caprine *KRTAP24-1*. These primers were synthesised by the Takara Biotechnology Company Limited (Dalian, China). Amplifications were performed in a 20 μL reaction containing the genomic DNA purified from a 1.2 mm punch of dried blood, 0.25 μM of each primer, 150 μM of each dNTP (Takara, Dalian, China), 2.5 mM Mg^2+^, 0.5 U of *Taq* DNA polymerase (Takara, Dalian, China), and 1× the PCR buffer supplied with the enzyme. The thermal profile consisted of an initial denaturation for 2 min at 94 °C, followed by 35 cycles of 94 °C for 30 s, 63 °C for 30 s, and 72 °C for 30 s, with a final extension of 5 min at 72 °C. Thermal cycling was undertaken in Bio-Rad S1000 thermal cyclers (Bio-Rad, Hercules, CA, USA).

For each amplicon, a separate 0.7 μL aliquot was added to 7 μL of loading dye (98% formamide, 10 mM EDTA, 0.025% bromophenol blue, 0.025% xylene cyanol). These samples were denatured at 95 °C for 5 min, then rapidly cooled on wet ice, prior to being loaded onto 16 × 18 cm, 12% acrylamide/bisacrylamide (37.5:1) (Bio-Rad, Hercules, CA, USA) gels. Electrophoresis was performed using Protean II xi cells (Bio-Rad, Hercules, CA, USA) for 22 h in 0.5 × TBE at 240 V and 17 °C, and the gels were stained using the method described by Byun et al. [20].

### 2.4. Sequencing of Allelic Variants and Sequence Analyses

For those amplicons that were deemed to be homozygous by PCR-SSCP analysis, DNA sequencing was performed directly and in both directions at the Beijing Genomics Institute, Beijing, China. For the variants that were typically rarer, and only found in a heterozygous form, DNA sequencing was undertaken using an approach described by Gong et al. [21]. In this approach, a band from the PCR-SSCP gel corresponding to the rare variant was excised, macerated, and then used to provide a DNA template for re-amplification with the original primers. After confirming that only this band had been amplified with a second round of SSCP analysis, this amplicon was sequenced as above. The BLAST algorithm was used to search the NCBI GenBank (http://www.ncbi.nlm.nih.gov/) databases for similar sequences.

DNAMAN version 5.2.10 (Lynnon BioSoft, Vaudreuil, QC, Canada) was used for the translation of open reading frames to amino acid sequences, the alignment of DNA sequences and amino acid sequences, and the creation of phylogenetic trees.

### 2.5. Statistical Analyses

The statistical analyses were performed using IBM SPSS Statistics version 24.0 (IBM, NY, USA). The General Linear Mixed-Effects Models (GLMMs) option was used to ascertain the effect of *KRTAP24-1* genotype on the measured cashmere traits. Due to the multiple comparisons undertaken in these models, a Bonferroni correction was applied. Sire and gender were found to affect (*p* < 0.05) all the cashmere fibre traits, and, accordingly, they were included in the models (as random and fixed factors, respectively). Only main effects were tested, and associations were considered significant at the 5% level.

## 3. Results

### 3.1. Identification of Caprine KRTAP24-1

A BLAST search of the Caprine Genome Assembly GCF_001704415.1 using a sheep *KRTAP24-1* sequence (JX112014) revealed a homologous region on goat chromosome 1, and an open reading frame of 774 bp was found at position 4878528_4879301 in the caprine genome sequence NC_022293.1. Eight previously described *KRTAPs* were identified upstream of this region [7], and in order from the centromere to the telomere, these were *KRTAP11-1*, *KRTAP7-1*, *KRTAP8-1*, *KRTAP8-2*, *KRTAP6-2*, *KRTAP20-2*, *KRTAP13-1*, and *KRTAP13-3* (Figure 1).

Four unique PCR-SSCP banding patterns (named *A*, *B*, *C*, and *D*) were detected for caprine *KRTAP24-1*. Either one or a combination of two different patterns was observed for each goat analyzed, these representing homozygous and heterozygous animals, respectively (Figure 2). DNA sequencing of the amplicons that produced these patterns revealed four unique nucleotide sequences, and while all of these sequences were different, they still had a similarity of over 99% with the putative *KRTAP24-1* sequence (NC_022293.1) in the caprine genome assembly.

Phylogenetic analysis revealed that these caprine sequences were different to all of the caprine sequences identified to date but were most closely related to the *KRTAP24-1* sequences from sheep and humans (Figure 3). This suggests that these newly identified sequences were caprine orthologous variants of *KRTAP24-1*. These variant sequences were deposited into GenBank with the accession numbers MG996011−MG996014.

The protein that might be encoded by caprine *KRTAP24-1* would comprise 257 amino acids and would contain 8.6–9.0 mol.% of cysteine, 15.6 mol.% of serine, and 7.0 mol.% of tyrosine.

### 3.2. Variation in Caprine KRTAP24-1

Nine single-nucleotide polymorphisms (SNPs) were detected across the caprine variant sequences. Of these SNPs, eight were within the coding region, and four of them would result in amino acid changes (Figure 4). It is notable that there was a C/T SNP located two nucleotides upstream from the predicted ATG codon. In the regions around this position and at c.618, variants *A* and C were identical, while *B* and *D* were identical. However, in the regions around c.252 and c.656 and the region spanning c.319 to c.417, *A* and *D* were identical, and *B* and *C* were identical (Figure 4).

### 3.3. Association between Variation in KRTAP24-1 and Cashmere Traits

Six genotypes (*AA*, *AB*, *BB*, *AC*, *AD*, and *BD*) were found in the cashmere goats investigated. Their frequencies were 32.06%, 47.06%, 17.65%, 1.18%, 0.88%, and 1.18% for *AA*, *AB*, *BB*, *AC*, *AD*, and *BD*, respectively. *AC, AD*, and *BD* were each detected at a frequency of less than 5%; hence, the association of these genotypes with cashmere traits was not investigated given this low frequency and potential for bias. Associations were accordingly only tested for the three common genotypes: *AA*, *AB*, and *BB*.

Genotype was found to have an effect on cashmere MFD. Cashmere produced by *AA* goats had the lowest MFD, while cashmere produced by *BB* goats had the highest MFD. Cashmere produced by *AB* heterozygote goats had an intermediate MFD, but it was significantly different to the MFDs of goats with the *AA* and *BB* genotypes (Table 1). No associations were found between the genotypes and cashmere weight or crimped fibre length.

## 4. Discussion

This study describes the identification of a new caprine KAP gene called *KRTAP24-1* that encodes an HS-KAP protein. The gene was clustered with eight previously described KAP genes including three HS- and five HGT-KAP genes on goat chromosome 1. It exhibited the highest similarity with the *KRTAP24-1* sequence from sheep and human when compared to any previously described HS-KAP sequences from goat, sheep, and human. It was concluded that the newly identified gene represents the orthologous caprine *KRTAP24-1*.

While the putative protein encoded by caprine *KRTAP24-1* could be assigned into the HS-KAP group, the caprine KAP24-1 protein would contain a much lower content of cysteine compared to any other HS-KAP. In contrast, this protein would have a relatively high content of serine and tyrosine. Cysteine residues are thought to form disulfide bonds that cross-link with the IFs [23], whereas tyrosine residues in HGT-KAPs may regulate the arrangement of IFs via cation–π interactions [24]. It has also been suggested that with ovine KAP11-1 and KAP13-3, the serine residues may be phosphorylated [25,26] and that phosphorylation might therefore affect the assembly and organisation of the keratins [27]. Having a lower content of cysteine and a higher content of serine and tyrosine is not common in HS-KAPs. This phenomenon has been described previously for KAP24-1 in other species (human and sheep) [16,17] and for KAP11-1 and KAP13-3 in sheep [25,26]. The functional significance of this unusual amino acid composition is unknown.

The amount of variation found in the caprine *KRTAP24-1* sequence is comparable to that reported for ovine *KRTAP24-1* [17]. It is also consistent with what has been observed with other *KRTAPs* in goats [7,8,28] and sheep [29,30]. There is some evidence that gene conversion events or non-reciprocal genetic exchange may have occurred with these *KRTAP24-1* sequences, as similarities are observed with what has been described to underpin the nucleotide variation observed in *KRTAP1-n* and *KRTAP15-1,* the suggestion being that it has been driven by the activity of *Chi*-like sequences [31,32]. *Chi* (crossover hotspot instigator, χ) is an octamer sequence (5′-GCTGGTGG-3′) associated with recombination hotspots in *E. coli* and is where recombination is initiated by double-strand DNA breaks [33]. Variants of this *Chi*-motif are suggested to have partial recombinogenic activity, and analysis of the *KRTAP24-1* sequences reveals a *Chi*-like sequence from c.380 to c.387 (5′-CCCCCAGC-3′, reverse complementary to 5-GCTGGGGG-3′). Together, this suggests that further research is needed into the evolutionary origins of the KAP genes.

The observation that there were more *AA* and *AB* genotypes and fewer *BB* genotypes in Longdong cashmere goats appears to be consistent with the association results obtained here. Genotypes *AA* and *AB* were found to be associated with finer cashmere fibre than genotype *BB*. This is consistent with this trait been selected for in Longdong cashmere goats, leading to *AA* and *AB* becoming more common in the population.

The KAP genes are clustered together by chromosomal region, and research in sheep has revealed that all of the KAP genes identified to date are polymorphic [2,4,5]. This raises the possibility that the associations described here for *KRTAP24-1* may be due to linkage with other *KRTAPs* clustered in the same chromosome region. Eight other *KRTAPs* have been found to be located upstream from *KRTAP24-1* on goat chromosome 1, and these are *KRTAP13-3, KRTAP13-1, KAP20-2, KRTAP6-2, KRTAP8-2, KRTAP8-1*, and *KRTAP11-1* [7]. The association detected for *KRTAP24-1* is similar to that reported for *KRTAP8-2* [9] but is different to that described for *KRTAP20-2* [7] and *KRTAP13-1* [10]. Given that *KRTAP24-1* is physically closer to *KRTAP20-2* and *KRTAP13-1* than *KRTAP8-2* on the chromosome (Figure 1), it would suggest that the association obtained here is not because of the effect of the other nearby *KRTAPs* but instead represents the independent effect of *KRTAP24-1.*

Variation in caprine *KRTAP24-1* may affect cashmere traits in different ways. Firstly, a number of the SNPs that were found were non-synonymous and would result in amino acid changes in the putative KAP24-1 protein. These amino acid changes may affect the protein structure, its property, or its interaction with IFs, and consequently affect fibre traits. There were three non-synonymous SNPs (c.301T/C, c.319G/A, and c.656T/C) when comparing variants *A* and *B* for which a difference in MFD was detected here. Two of these SNPs are worthy of attention. SNP c.301T/C would result in gain or loss of cysteine, whereas SNP c.319G/A would result in gain or loss of glycine. Cysteine is usually the first-limiting amino acid for wool or cashmere fibre synthesis [34] and is essential for the formation of disulphide bonds between KAPs and IFs, whereas glycine is a small residue that may make the KAP protein more flexible and thus better able to form a compact structure with IFs. It is logical that the gain of cysteine and glycine for variant *A* compared to *B* may favour KAP24-1 forming a more compact structure with the IFs, and thus lead to a finer fibre, this being consistent with the association results. However, as described below, these SNPs may simply be linked to other genetic variation that might affect the expression of *KRTAP24-1*.

Synonymous SNPs can affect gene expression or protein structure. It has been suggested that “silent” variation can affect in vivo protein folding and, consequently, protein function by creating less common codons in mRNA, which slow the rate of translation and protein formation, leading to structural changes [35,36].

Finally, SNPs near to ATG start codons may influence the efficacy of translation and, thus, gene expression. One SNP was located two base pairs upstream of the start codon, with the nucleotide being a cytosine (C) in variants *A* and *C* but being substituted to a thymine (T) in variants *B* and *D* (Figure 4). In eukaryotes, the ribosome requires a specific sequence on a mRNA molecule as the translational start site, with gcc(A/G)ccAUGG from c.−6 to c.4 being proposed as the optimal sequence for initiation of translation [37], where a lowercase letter denotes the most common base at a position where the base can, nevertheless, vary. Uppercase letters indicate highly conserved bases, and the typical AUG initiator codon is underlined. While a cytosine at c.−2 is not absolutely conserved, it can contribute to the overall strength of ribosome binding [38]. The substitution of cytosine (C) to thymine (T) seen in variants *B* and *D* may therefore affect the translation of the KRTAP24-1 mRNA, and, hence, less KAP24-1 protein is produced to cross-link the IFs. This may lead to a less compact fibre being produced and, hence, an association with high MFD, as was described for variant *B* when compared to variant *A*.

Further investigation is needed to reveal how variation in *KRTAP24-1* may affect cashmere traits. Nevertheless, the association found in the study suggests that *KRTAP24-1* could be a gene marker for fine cashmere fibre.

## 5. Conclusions

This study identified the caprine KAP24-1 gene and revealed variation in the gene. Variation in the caprine KAP24-1 gene was found to affect cashmere fibre diameter. These results may be useful in the future development of breeding programs based on improving cashmere fibre diameter.

## Figures and Tables

**Figure 1 animals-09-00015-f001:**
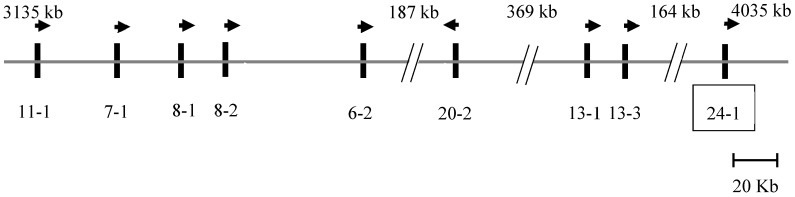
Location of *KTRTAPs* on caprine chromosome 1. The putative *KRTAP24-1* (boxed) is clustered with eight previously identified *KRTAPs* [7]. The vertical bars represent the genes, and the arrows indicate the direction of transcription of these genes. The names of the genes are written below the bars (e.g., 11-1 represents *KRTAP11-1*). The spacing of the genes is only approximate and is based on the Caprine Genome Assembly. The nucleotide coordinates are given relative to GCF_001704415.1 [19] and are approximate. A pair of diagonal lines indicate that the spacing is not proportional to the length of the sequence, with the distance being shown above.

**Figure 2 animals-09-00015-f002:**
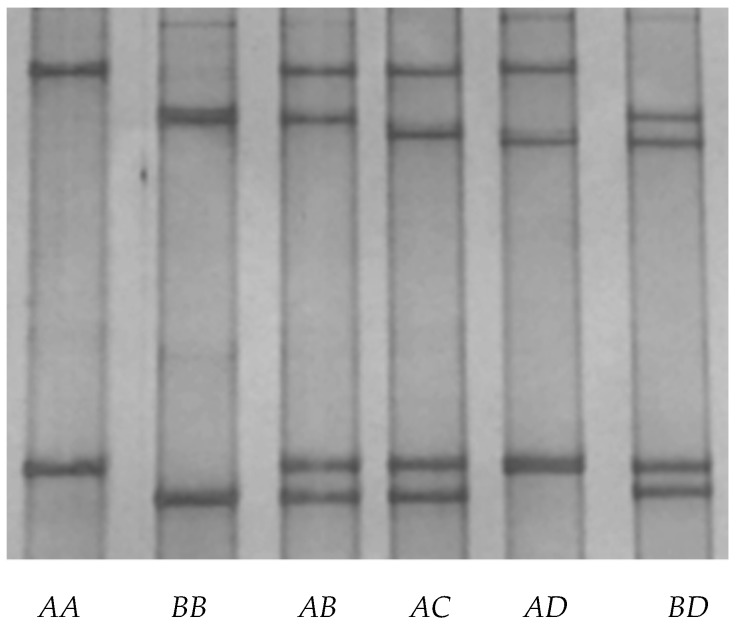
Variation in the caprine KAP24-1 gene detected by PCR single-strand conformation polymorphism analysis. Four unique banding patterns representing four variants (*A* to *D*) were observed in either homozygous or heterozygous forms.

**Figure 3 animals-09-00015-f003:**
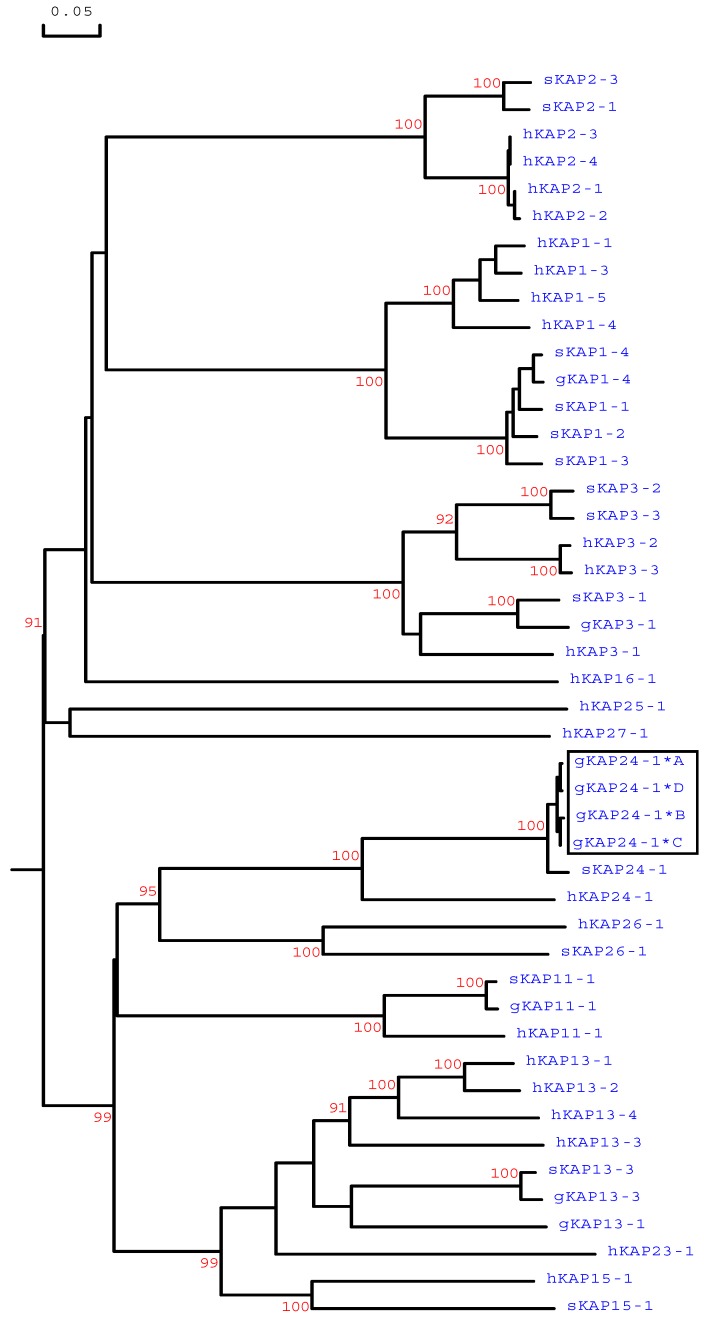
Phylogenetic tree of high-sulphur (HS)-KAPs identified in sheep, humans, and goats. Amino acid sequences or predicted amino acid sequences were used to construct the tree. In the tree, the numbers printed at the forks indicate the bootstrap confidence values. Only those values equal to or higher than 90% are shown. The sheep sequences are indicated with “s”, the human sequences are indicated with “h”, and the caprine KAPs are indicated with a prefix “g”. The four newly identified goat KAP24-1 sequences are shown in a box, and the GenBank/EMBL accession numbers for other HS-KAPs are JN012101.1 (gKAP1-4), NM_001285774 (gKAP3-1), NM_001285767.1 (gKAP11-1), AY510115 (gKAP13-1), JX426138 (gKAP13-3), X01610 (sKAP1-1 and sKAP1-4), HQ897973 (sKAP1-2), X02925 (sKAP1-3), P02443 (sKAP2-1), P02441 (sKAP2-3), P02446 (sKAP3-1), P02444 (sKAP3-2), P02445 (sKAP3-3), HQ595347 (sKAP11-1), JN377429 (sKAP13-3), KX817979 (sKAP15-1), JX112014 (sKAP24-1), KX644903 (sKAP26-1), NM_030967.2 (hKAP1-1), NM_030966.1 (hKAP1-3), NM_001257305.1 (hKAP1-4), NM_031957.1 (hKAP1-5), NM_001123387.1 (hKAP2-1), NM_033032.2 (hKAP2-2), NM_001165252.1 (hKAP2-3), NM_033184.3 (hKAP2-4), NM_031958.1 (hKAP3-1), NM_031959.2 (hKAP3-2), NM_033185.2 (hKAP3-3), NM_175858.2 (hKAP11-1), NM_181599.2 (hKAP13-1), NM_181621.3 (hKAP13-2), NM_181622.1 (hKAP13-3), NM_181600.1 (hKAP13-4), NM_181623.1 (hKAP15-1), NM_001146182.1 (hKAP16-1), NM_181624.1 (hKAP23-1), NM_001085455.2 (hKAP24-1), NM_001128598.1 (hKAP25-1), NM_203405.1 (hKAP26-1), and NM_001077711.1 (hKAP27-1).

**Figure 4 animals-09-00015-f004:**
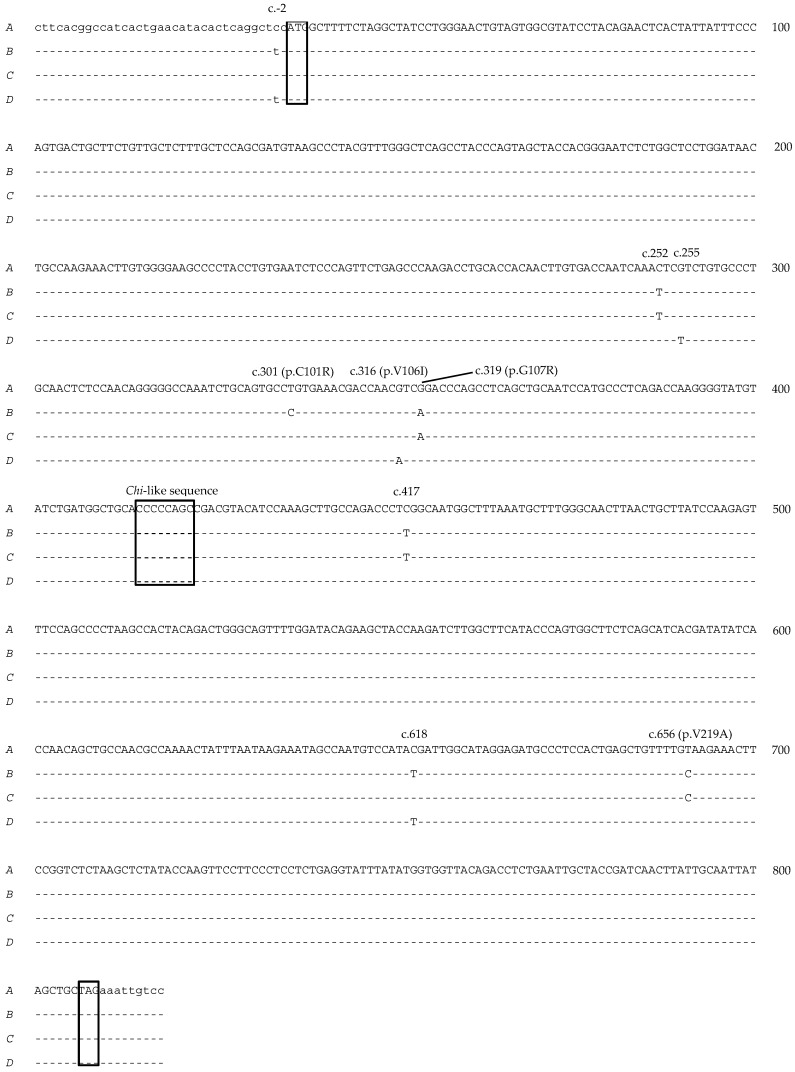
Nucleotide sequences of caprine *KRTAP24-1* variants *A–D*. The sequences exclude the PCR primer binding regions. Nucleotides in the coding region are shown in uppercase, while those outside the coding region are in lowercase. A putative *Chi*-like sequence (c.380 to c.387) and the predicted ATG start codon and TAG stop codon are boxed. Dashes signify homology with the *KRTAP24-1* variant *A* sequence. The single-nucleotide polymorphism (SNP) positions are shown above the sequences, and those that would result in amino acid changes are indicated. The numbering of nucleotides and amino acids follows the HGVS nomenclature guidelines [22].

**Table 1 animals-09-00015-t001:** The effect of *KRTAP24-1* genotype on various cashmere fibre traits (mean ± SE) ^1^ in Longdong cashmere goats.

Cashmere Trait (Unit)	Mean ± SE			*p* Value
	*AA* (*n* = 109)	*AB* (*n* = 160)	*BB* (*n* = 60)	
Cashmere weight (g)	414 ± 4.6	418 ± 3.8	411 ± 5.4	0.457
Mean fibre diameter (μm)	13.4 ± 0.04 ^C^	13.6 ± 0.04 ^B^	13.8 ± 0.05 ^A^	<0.001
Crimped fibre length (cm)	4.2 ± 0.05	4.3 ± 0.04	4.2 ± 0.06	0.301

^1^ Estimated marginal means and standard errors (SEs) derived from General Linear Models. In these models, “sire” was fitted as a random factor and “gender” as a fixed factor. A Bonferroni correction was used to correct for multiple comparisons made in the models. Means within rows that do not share a superscript letter (e.g., A, B, or C) are different at *p* < 0.01 and shown in bold.

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
