# Peer review of "Variation in the Caprine KAP24-1 Gene Affects Cashmere Fibre Diameter"

_animals, 2019, doi:10.3390/ani9010015_

Round 1
Reviewer 1 Report
In the present study, authors studied genetic variations in the Keratin Associated Protein 24-1 gene (KRTAP24-1) in the population of cashmire goat Longdong breed. The fleece of cashmere goats comprises cashmere that is the fibre produced by the secondary fibre follicles. It is also known as the keratin intermediate filaments. They are embedded in an interfilamentous matrix, consisting of fleece hair or fleece keratin-associated proteins (KRTAP), which are essential for the formation of a rigid and resistant fleece shaft through their extensive disulfide bond cross-linking with abundant cysteine residues of fleece keratins. The matrix proteins include the high-sulfur and high-glycine-tyrosine keratins. Genetic variations in this gene at the post-translational modification sites for instance are expected to affect the cashmere fleece trait. This is of economical important for the country/industry/producers.
From a technical perspective, the study was well conducted.
Nomenclature of genes has been used accordingly. There are however some aspects that require to be addressed to better reflect the work and to reflect the scope of the study.
Major Compulsory Revisions
Lines 297-298. The population used for the study included the progeny of ten sire families. Were these 10 families geographically well distributed and representative of the population of Longdong cashmere goats? Please provide information on the proportion of these 340 Longdong-type animals compared to the existing Cashmire goat population in China. Was this proportion representative of the existing population? If this is not the case, please qualify and include some ideas in your discussion to find out if more genetic variation can be expected in the gene. Also indicate that the PCR-SSCP is an exploratory technique and that sequencing could possibly provide more precision in the identification of genetic variations in the population.
Authors are using “variation sequences” which introduce some confusion. Were they genetic variations (e.g. SNP) or DNA sequences ? alternatively, authors can use distinct DNA sequences for the KRTAP24-1 gene were observed within the population. Or: Distinct DNA sequences were identified by sequencing the KRTAP24-1 gene in the population of cashmire goats.
There is a confusion in the usage of the terms families or homologous gene, whereas paralogous, orthologous, or similar sequence should rather be used. Authors need to revise them according to the nomenclature; refer to “Information about “Orthologs, Paralogs, and Evolutionary Genomics” that can be found in the journal Annual Review of Genetics, vol 39:309-338. Orthologous and paralogous genes are different types of homologous genes. Homologous genes are two or more genes that descend from a common ancestral DNA sequence. An example of homologous genes are the genetic codes underlying genes that produce the hemoglobin and myoglobin proteins. Both retain similar features but are utilized in different manners. Myoglobin arose in this ancestral species as a paralogous gene to hemoglobin; a mutation in the hemoglobin gene during a duplication event resulted in a separate myoglobin gene that carries out a new, yet similar, function. The eight described KRTAPs are paralagous genes. Some can be combined to define the trait of cashmire fleece. These traits, which were passed down from their last common ancestor, have adaptive pressures that may lead to variations within the gene. The point or event in evolutionary history that accounts for the DNA sequence variation within the gene determines whether the homologous genes are considered ‘ortho’ or ‘para’. This study is about using sequence homology for the identification of the orthologous caprine version of KAP24-1 and to define DNA sequence variation within the gene that would affect the fibre diameter trait of the cashmire fleece.
Minor revisions.
Revise punctuation, which was often not appropriately used: as example
Line 25: no comma between the subject and the verb; Line 34: remove comma before “and” because no comma is needed between an enumeration of 2 items; line 49: change comma to a new sentence “. Therefore, “; etc. revise the entire text thoroughly.
Lines 18-19: Rewrite the sentence in to: The human and sheep orthologous gene encoding the high sulphur (HS)-KAP24-1 is now described for the goat species.
Please refer to manuscript entitled “Information about “Orthologs, Paralogs, and Evolutionary Genomics” that can be found in the journal Annual Review of Genetics, vol 39:309-338.
The sentence at lines 78-80 should also be revised accordingly. Elsewhere in the text also.
Line 26: remove the word “notionally” and elsewhere. Not sure that it is appropriate in this context. There is no “theory” presented in this manuscript.
Line 29: Change “reaction-single” to “reaction using the single”
Line 30: replace “banding patterns” to “distinct pattern of DNA bands on gel electrophoresis”
Line 31; The 340 are not truly a population because some are progeny of the sire. It would be more appropriate to write “ten families of Longdong cashmere goats totalizing 340 goats”
Lines 31-33: rewrite the sentence such as: The genetic variations in the KRTAP24-1 gene were similar to the variations observed in the orthologous sheep or human gene, …
Line 34: remove “typical”
Line 49: Change “understood” to known
Lines 61-62, 66-67, and elsewhere in the paragraph starting line 63: the term paralogous would be appropriate in the context.
Lines 80-81: This sentence clearly defines the goal of the study and should be also be used in the abstract.
Line 122: the word homologous refers to “A gene related to a second gene by descent from a common ancestral DNA sequence.” It seems that you mean “similar”.
Lines 167-168: change “variant of caprine” to “caprine orthologous”
Line 168-169: You’re using variations (instead of sequences). Were they variations (SNP) or sequences?
Lines 170, 234: the meaning of “notional” is not clear in this context. It is very unusual, change for another word.
Line 269: do not use comma when enumerating only two parts. If the sentence is too long, make it short, i.e. starting a new sentence instead of using a comma. Comma is used when enumerating 3 or more items.
Line 272: you identified the “caprine orthologous gene” of the human, goat, and sheep
KRTAP24-1 genes. Use this meaning elewhere in the text as well.
Lines 280-281: remove both comma. This is not an enumeration. A second sentence should start at the third comma (This phenomenon was described….) Reducing the length of the sentence is easier for the reader.
Lines 284-285. Make two sentences of this.
Lines 286-287: “not scientifically sounds: “Little is understood of how this sequence variation has come about.” Sounds like an oral discussion which should not be found in a scientific manuscript.
Author Response
Reviewer 1
Comments and Suggestions for Authors
In the present study, authors studied genetic variations in the Keratin Associated Protein 24-1 gene (KRTAP24-1) in the population of Cashmere goat Longdong breed. The fleece of cashmere goats comprises cashmere that is the fibre produced by the secondary fibre follicles. It is also known as the keratin intermediate filaments. They are embedded in an interfilamentous matrix, consisting of fleece hair or fleece keratin-associated proteins (KRTAP), which are essential for the formation of a rigid and resistant fleece shaft through their extensive disulphide bond cross-linking with abundant cysteine residues of fleece keratins. The matrix proteins include the high-sulphur and high-glycine-tyrosine keratins. Genetic variations in this gene at the post-translational modification sites for instance are expected to affect the cashmere fleece trait. This is of economical important for the country/industry/producers.
From a technical perspective, the study was well conducted.
Nomenclature of genes has been used accordingly. There are however some aspects that require to be addressed to better reflect the work and to reflect the scope of the study.
Major Compulsory Revisions
Lines 297-298. The population used for the study included the progeny of ten sire families. Were these 10 families geographically well distributed and representative of the population of Longdong cashmere goats? Please provide information on the proportion of these 340 Longdong-type animals compared to the existing Cashmere goat population in China. Was this proportion representative of the existing population? If this is not the case, please qualify and include some ideas in your discussion to find out if more genetic variation can be expected in the gene. Also indicate that the PCR-SSCP is an exploratory technique and that sequencing could possibly provide more precision in the identification of genetic variations in the population.
This was never intended to be a population study. Few Chinese populations of cashmere goats have accurately recorded sire information, any documentary evidence of in/out breeding having occurred, or any detailed and precisely measured phenotypic data available. These are the reasons why these 340 Longdong goats were chosen. The goats are most certainly NOT representative of the wider population as they are specifically of the Longdong type. In contrast there are separate cashmere breeds that include the Changthangi (Kashmir Pashmina) goat, the Liaoning goat, the Inner Mongolian cashmere goat and the Hexi goat. In a population study it would be fair to say that we should have studied these populations too. In that respect, you might also expect to find further variation in the gene, as might also be expected if one was to look at other goat breeds globally.
Authors are using “variation sequences” which introduce some confusion. Were they genetic variations (e.g. SNP) or DNA sequences? alternatively, authors can use distinct DNA sequences for the KRTAP24-1 gene were observed within the population. Or: Distinct DNA sequences were identified by sequencing the KRTAP24-1 gene in the population of cashmere goats.
These were most certainly DNA sequences, which were identified initially using PCR-SSCP to screen these goats. The DNA sequences contain the nine SNPs that we describe. It is conceivable that other KRTAP24-1 sequences sill be revealed upon sequencing of this gene from more goats, and goats from both from China and other countries.
There is a confusion in the usage of the terms families or homologous gene, whereas paralogous, orthologous, or similar sequence should rather be used. Authors need to revise them according to the nomenclature; refer to “Information about “Orthologs, Paralogs, and Evolutionary Genomics” that can be found in the journal Annual Review of Genetics, vol 39:309-338. Orthologous and paralogous genes are different types of homologous genes. Homologous genes are two or more genes that descend from a common ancestral DNA sequence. An example of homologous genes are the genetic codes underlying genes that produce the haemoglobin and myoglobin proteins. Both retain similar features but are utilized in different manners. Myoglobin arose in this ancestral species as a paralogous gene to haemoglobin; a mutation in the haemoglobin gene during a duplication event resulted in a separate myoglobin gene that carries out a new, yet similar, function. The eight described KRTAPs are paralogous genes. Some can be combined to define the trait of cashmere fleece. These traits, which were passed down from their last common ancestor, have adaptive pressures that may lead to variations within the gene. The point or event in evolutionary history that accounts for the DNA sequence variation within the gene determines whether the homologous genes are considered ‘ortho’ or ‘para’. This study is about using sequence homology for the identification of the orthologous caprine version of KAP24-1 and to define DNA sequence variation within the gene that would affect the fibre diameter trait of the cashmere fleece.
See changes to text.
Minor revisions
Revise punctuation, which was often not appropriately used: as example
Line 25: no comma between the subject and the verb; Line 34: remove comma before “and” because no comma is needed between an enumeration of 2 items; line 49: change comma to a new sentence “. Therefore, “; etc. revise the entire text thoroughly.
All corrected.
Lines 18-19: Rewrite the sentence in to: The human and sheep orthologous gene encoding the high sulphur (HS)-KAP24-1 is now described for the goat species.
Corrected.
Please refer to manuscript entitled “Information about “Orthologs, Paralogs, and Evolutionary Genomics” that can be found in the journal Annual Review of Genetics, vol 39:309-338.
The sentence at lines 78-80 should also be revised accordingly. Elsewhere in the text also.
Corrected.
Line 26: remove the word “notionally” and elsewhere. Not sure that it is appropriate in this context. There is no “theory” presented in this manuscript.
Corrected.
Line 29: Change “reaction-single” to “reaction using the single”
Wording revised.
Line 30: replace “banding patterns” to “distinct pattern of DNA bands on gel electrophoresis”
Wording revised.
Line 31; The 340 are not truly a population because some are progeny of the sire. It would be more appropriate to write “ten families of Longdong cashmere goats totalizing 340 goats”
It isn’t claimed that they are a population or representative of a population. The methods state clearly that ‘A total of 340 Longdong cashmere goats that were the progeny of ten unrelated sires were investigated.’
Lines 31-33: rewrite the sentence such as: The genetic variations in the KRTAP24-1 gene were similar to the variations observed in the orthologous sheep or human gene, …
Wording changed.
Line 34: remove “typical”
Removed.
Line 49: Change “understood” to known
Changed.
Lines 61-62, 66-67, and elsewhere in the paragraph starting line 63: the term paralogous would be appropriate in the context.
Word inserted.
Lines 80-81: This sentence clearly defines the goal of the study and should be also be used in the abstract.
Has been inserted in the abstract.
Line 122: the word homologous refers to “A gene related to a second gene by descent from a common ancestral DNA sequence.” It seems that you mean “similar”.
Changed to similar.
Lines 167-168: change “variant of caprine” to “caprine orthologous”
Changed.
Line 168-169: You’re using variations (instead of sequences). Were they variations (SNP) or sequences?
They were sequences, but they contained nine SNPs.
Lines 170, 234: the meaning of “notional” is not clear in this context. It is very unusual, change for another word.
We have no evidence that the gene is expressed, hence use of the word notional. We have change the wording to ‘protein that might be expressed’, or ‘predicted’, as required.
Line 269: do not use comma when enumerating only two parts. If the sentence is too long, make it short, i.e. starting a new sentence instead of using a comma. Comma is used when enumerating 3 or more items.
Sentence has been split in two.
Line 272: you identified the “caprine orthologous gene” of the human, goat, and sheep KRTAP24-1 genes. Use this meaning elsewhere in the text as well.
Wording changed.
Lines 280-281: remove both comma. This is not an enumeration. A second sentence should start at the third comma (This phenomenon was described….) Reducing the length of the sentence is easier for the reader.
Sentence has been split in two.
Lines 284-285. Make two sentences of this.
Sentence has been split in two.
Lines 286-287: “not scientifically sounds: “Little is understood of how this sequence variation has come about.” Sounds like an oral discussion which should not be found in a scientific manuscript.
Sentence deleted.

Reviewer 2 Report
The manuscript describes the characterisation of variation in KAP24-1 gene in goats, which transcribes proteins that are important structural proteins for cashmere fibres. PCR_SSCP is used to identify four different sequences with unique banding patterns. 340 Longdong goats from China are screened in the study and fibre phenotypes are associated with the observed sequence variants, to determine the potential suitability of KAP24-1 as a genetic marker for improving cashmere fibre diameter.
The manuscript is well written and clearly describes the findings. I only have some minor line changes.
Line 18 Remove ‘the’ before ‘high sulphur’ and also check this throughout the manuscript
Line 78 Change ‘As of date’ to ‘To date’
Line 286 Change ‘of’ to ‘as to’
Line 287 Change “has come about” to “has arisen”
Line 288 Change ‘in this respect’ to “similarly”
Line 300 End sentence at ‘then genotype BB’ and start new sentence and change to “This is consistent with selection for finer cashmere fibre in Longdong cashmere goats…”
Line 302 Change to ‘KAP genes are cluster together by chromosomal region and research in sheep has….’
Line 329 Change ‘structure’ to ‘structural’
Line 331 insert ‘of’ before ‘the start codon’
Author Response
Reviewer 2
Comments and Suggestions for Authors
The manuscript describes the characterisation of variation in KAP24-1 gene in goats, which transcribes proteins that are important structural proteins for cashmere fibres. PCR_SSCP is used to identify four different sequences with unique banding patterns. 340 Longdong goats from China are screened in the study and fibre phenotypes are associated with the observed sequence variants, to determine the potential suitability of KAP24-1 as a genetic marker for improving cashmere fibre diameter.
The manuscript is well written and clearly describes the findings. I only have some minor line changes.
Line 18 Remove ‘the’ before ‘high sulphur’ and also check this throughout the manuscript
Removed.
Line 78 Change ‘As of date’ to ‘To date’
Changed.
Line 286 Change ‘of’ to ‘as to’
Sentence re-written.
Line 287 Change “has come about” to “has arisen”
Sentence re-written.
Line 288 Change ‘in this respect’ to “similarly”
Changed.
Line 300 End sentence at ‘then genotype BB’ and start new sentence and change to “This is consistent with selection for finer cashmere fibre in Longdong cashmere goats…”
Changed.
Line 302 Change to ‘KAP genes are cluster together by chromosomal region and research in sheep has….’
Changed.
Line 329 Change ‘structure’ to ‘structural’
Changed.
Line 331 insert ‘of’ before ‘the start codon’
‘Of’ added.
